# Data navigation on the ENCODE portal

Meenakshi S. Kagda ®, Bonita Lam, Casey Litton, Corinn Small, Cricket A. Sloan ®, Emma Spragins, Forrest Tanaka, Ian Whaling, Idan Gabdank ®, Ingrid Youngworth, J. Seth Strattan, Jason Hilton, Jennifer Jou, Jessica Au ®, Jin-Wook Lee, Kalina Andreeva, Keenan Graham, Khine Lin ®, Matt Simison, Otto Jolanki ®, Paul Sud ®, Pedro Assis, Philip Adenekan, Stuart Miyasato, Weiwei Zhong, Yunhai Luo ®, Zachary Myers, J. Michael Cherry ® & Benjamin C. Hitz ® ✉

Spanning two decades, the collaborative ENCODE project aims to identify all the functional elements within human and mouse genomes. To best serve the scientific community, the comprehensive ENCODE data including results from 23,000+ functional genomics experiments, 800+ functional elements characterization experiments and 60,000+ results from integrative computational analyses are available on an open-access data-portal (https://www.encodeproject.org/). The final phase of the project includes data from several novel assays aimed at characterization and validation of genomic elements. In addition to developing and maintaining the data portal, the Data Coordination Center (DCC) implemented and utilised uniform processing pipelines to generate uniformly processed data. Here we report recent updates to the data portal including a redesigned home page, an improved search interface, new custom-designed pages highlighting biologically related datasets and an enhanced cart interface for data visualisation plus user-friendly data download options. A summary of data generated using uniform processing pipelines is also provided.

The ENCODE data portal (https://www.encodeproject.org) hosts a rich and diverse collection of biological datasets that are a result of a massive NHGRI-funded research consortium-level effort that lasted for more than two decades. The project was divided into four phases, and the final phase (ENCODE 4) came to a completion in December 2022. During these different phases, research labs across the globe joined forces to handle an enormous task of identifying all functional DNA elements present in the human genome by utilising several newer research approaches and cutting-edge techniques in the fields of Genetic Engineering, Genomics, Bioinformatics and Machine Learning. The raw sequencing data generated by the participating labs and the processed results generated by utilising uniform processing pipelines were deposited at a central repository and a database that is currently hosted at the ENCODE Data Portal. The portal not only facilitates cross-collaboration within the consortium but also serves as a rich open-source community resource for millions of scientists across the globe.

The current ENCODE data portal was first released in August 2014[1] and has since been expanded to accommodate a variety of consortium requirements. The database currently hosts more than 1.3 PB worth of genomics data with an average traffic of ~25,000 unique users per month worldwide (a total of ~1.3 M users per year). The portal provides rich metadata for each dataset, along with concomitant documentation. In addition, the user experience is enhanced by an intuitive interface that allows researchers to (a) easily navigate the data portal, (b) sub-select datasets of their interest using faceted search[2], (c) visualise the outputs of multiple genomics datasets by using an in-built Valis genome browser [https://github.com/VALIS-software/VALIS-frontend] and (d) download relevant data using different file selection options. The data portal can also be queried using an API (Application Programming Interface); this ensures that the data can be downloaded using computational methods in addition to web browser-based options.

Department of Genetics, Stanford University School of Medicine, Stanford, CA, USA. ✉e-mail: hitz@stanford.edu

Subsequent to our prior publication in 2020[3], new features have been added to the website. These features have been aimed to improve aspects of user experience as well as to accommodate newer data types. For example, we have completely re-designed the home page such that it provides quick links to several important datasets and collection pages using a single click on the various homepage shortcut cards. We have improved the cart functionality such that it allows creation of curated groupings in several different carts, name them as needed, provide useful descriptions, visualise the bed and bigWig files on a genome browser, filter the results based on both file properties as well as dataset properties within the cart and ultimately download specific file subsets from within the carts. There is also a new Valis genome browser interface called the Encyclopaedia Browser that allows simultaneous visualisation of results from genome annotations that were generated using integrative analysis methods constituting the Encyclopaedia of DNA elements.

In the following sections, we provide a detailed description of (1) an overview of the various types of ENCODE dataset types, (2) ways to navigate the different dataset types on the portal, (3) the new home page, (4) improved search function capabilities and (5) a description of the newly implemented cart functionality. We also provide a detailed description of various ENCODE Series and Collections in the Supplemental data.

## Results

### Summary of ENCODE datasets

As of March 2024, there are ~106 K released ENCODE-generated datasets that are publicly available* (See notes in Supplemental data). These datasets can be further categorised into several different types. The six main sub-categories are: (1) functional genomics experiments, (2) functional characterisation experiments including transgenic enhancer experiments, (3) single cell experiments, (4) series, (5) computational and integrative products (annotation datasets) and (6) publications and corresponding data objects.

In our encodeD database system [https://github.com/ENCODE-DCC/encoded], each of these data types (and other relevant metadata) are captured in schema-defined objects. A JSON schema is used to keep the properties and links of the objects consistent. In total, we have 118 types of objects. A full list, along with their properties, can be found on the ENCODE profiles page at: https://www.encodeproject.org/profiles/.

A "collection" page of several objects of the same type can be viewed on the portal in at least two formats: search view and report view. For example, all the experiment objects on the data portal can be viewed on these two rendered UI pages: https://www.encodeproject.org/search/?type=Experiment and https://www.encodeproject.org/report/?type=Experiment. For some key dataset objects, a matrix view is also available. The different views available for each of the objects have several predefined facets located on the left section of the page, which allow further refining of the search results based on relevant metadata. The facets also serve as important pointers about the variety of data available for selection. The facets have been grouped into relevant categories such that each of these grouped facets could be expanded or collapsed as needed. For a more detailed description on using facets and query building, along with the ENCODE resource and data objects, the readers can refer to ENCODE help pages (https://www.encodeproject.org/help/getting-started/) and our prior publications[2,4].

### Functional genomics experiments

The functional genomics experiments on the ENCODE portal are referred to as "Experiment" objects (Fig. 1). There are a total of 23,330 released experiments (http://tinyurl.com/ENCODE-experiments) hosted on the ENCODE portal. The majority of these datasets are a result of various phases of ENCODE awards (ENCODE 2, 3, 4, ENCODE2-mouse), and the remaining experiments were either imported from other allied

consortia efforts or directly submitted under those awards, such as modENCODE and modERN[5], Genomics of Gene regulation (GGR)[6] and Roadmap Epigenomics Mapping Consortium (REMC)[7]. The portal also hosts a few "community" datasets generated by laboratories not associated with one of the above-mentioned consortia. The breadth of genomics datasets (Fig. 1) generated by the ENCODE funded labs is evident by the variety of the diverse assay types (Fig. 2). The main assay types include DNA binding assays, transcription assays, DNA accessibility assays, single cell assays, 3D chromatin structure assays, RNA binding assays, DNA methylation assays and assays that decipher the RNA structure. A full breakdown of all the genomics assay types can be found in Fig. 2.

Most ENCODE-funded labs conducted experiments on human and mice samples, with a few exceptions that were performed in cell-free or in vitro systems (Fig. 1B, C). The experiments from modERN and modENCODE projects have data from three different Drosophila species (with a majority being from *Drosophila melanogaster*) as well as from *Caenorhabditis elegans*. In addition, these assays have been performed on different biosample types, including tissues, cell lines, primary cells, in vitro differentiated cells and organoids (Fig. 1C).

The default landing page for the Experiment search (http://tinyurl.com/exp-search) and report views don't display (i.e., hide) control assays and assays performed on perturbed biosamples. Using the facets, these experiments can be viewed or hidden using the "Hide control experiments" and "Perturbation" options, respectively. The latter facet helps filter assays that were performed on modified (treated or genetically altered) samples created for the purpose of testing the effects of the perturbation on the biosample.

### Uniformly processed data

In addition to storing raw fastq files and metadata that is relevant to each of the experiments, the experiment objects also contain processed data files. Experiments typically either host only lab-processed data files or only ENCODE pipeline-based uniformly processed data files. In some experiments, though, there could also be a combination of both types. The uniformly processed pipeline data files are available for a key set of assays that make up the bulk of the ENCODE experimental corpus, including: Histone ChIP-seq, TF ChIP-seq, MINT ChIP-seq, ATAC-seq, DNase-seq, RNA-seq, micro RNA-seq, long read RNA-seq, HiC and WGBS assays (https://www.encodeproject.org/data-analysis/). The software implementation of uniform processing pipelines was done as a collaborative effort within the ENCODE consortium, and the code for all the pipelines is freely available for public use on GitHub (see code availability section for links). More details on the uniform processing pipelines can be found in a separate publication[8]. In addition to displaying (for every dataset available on the portal) the relation between the various processed files generated by an execution of a pipeline run, the file association graphs also display useful quality metrics and summary statistics[9].

### Functional characterisation experiments

The Functional Characterisation experiments include various assays that were performed as validations of the activity of predicted functional DNA elements (http://tinyurl.com/Functional-characterization). Many of these predicted DNA elements were selected for analysis by utilising the processed data outputs of functional genomics experiments. These assays include (A) sequencing-based methods and (B) imaging-based methods.

A total of 737 sequencing-based datasets (including relevant controls) exist on the ENCODE portal that are classified as the Functional Characterisation Experiments. CRISPR screens are grouped into Functional Characterisation Series to demonstrate the relation between several different readouts. As shown in Fig. 3, the various assays performed under this umbrella include several flavours of

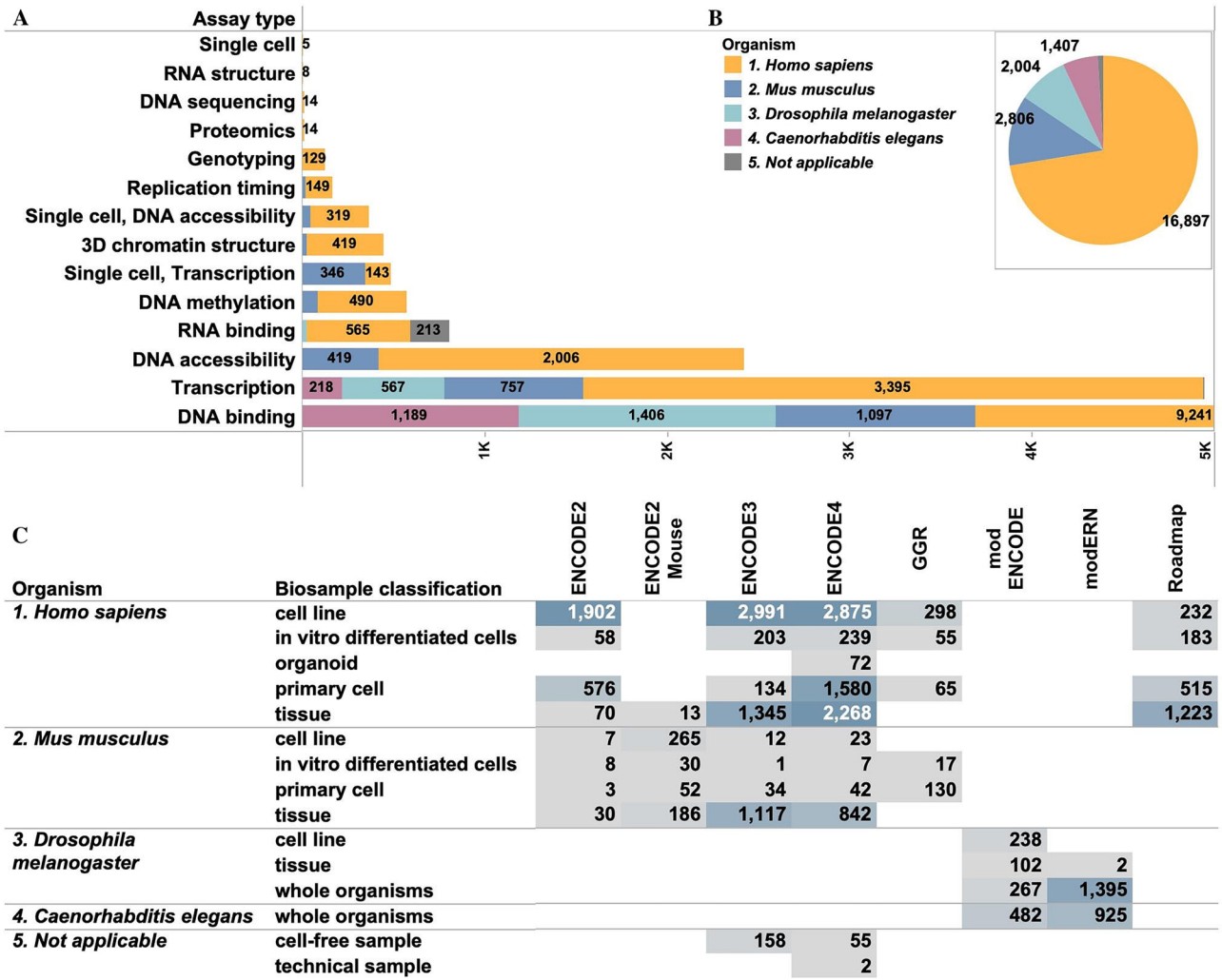

**Fig. 1 | Overview of different ENCODE functional genomics experiments.** The diversity and numbers of various ENCODE functional genomics experiments classified by (**A**) Assay type, (**B**) Biosample organism, (**C**) Biosample classification and awards are presented. As seen in Fig 1A, the highest number of ENCODE datasets include DNA binding based assays followed by Transcription based assays. Fig 1A and 1B further demonstrates that the maximum number of assays within the ENCODE corpus are composed of samples derived from *Homo sapiens* followed by *Mus musculus*. Fig 1C classifies the functional genomics experiments based on the different ENCODE phases as well as based on the biosample classification and organism type. As demonstrated in the figure, most functional genomics experiments within the ENCODE4 phase are encompassed of cell line based assays followed by tissue samples within the *Homo sapiens* category. In contrast, most of the *Mus musculus* based assays within the ENCODE3 and ENCODE4 phases have been performed on tissue samples.

CRISPR screens[10] as well as high-throughput Reporter Assays such as MPRA[11] and STARR-seq[12]. This dataset type also includes single-cell assays performed in cell lines perturbed using CRISPR machinery, sometimes also referred to as Perturb-seq[13] or SPEAR-ATAC[14]. Most of the assays that belong to this category were conducted in human biosamples, along with a few from mouse cell lines (Fig. 3).

There are 311 imaging-based transgenic enhancer reporter experiments (http://tinyurl.com/Transgenic-Enhancer) performed in mice embryos (E11.5 days). The construct tested in these datasets consists of an enhancer element cloned upstream of a promoter sequence, and the element activity is recorded using a beta-galactosidase reporter[15]. All the microscopic image outputs are provided by the ENCODE portal as attached viewable characterisation objects on the respective datasets. The data can also be viewed on the Vista enhancer browser[16] (http://enhancer.lbl.gov/).

## Single cell datasets
Single-cell-based genomics assays can be found on the Single Cell page (http://tinyurl.com/Single-cells). This page is further divided into three different tabs to separately support high-throughput, perturbed high-throughput and low-throughput assays. The high-throughput tab is further divided into human and mouse with interactive body maps. Both human and mouse tabs contain datasets for snATAC-seq and scRNA-seq. In addition to these two assay types, the mouse tab also has long-read scRNA-seq assays. The Perturbed high-throughput tab contains links to the perturb-seq-like functional characterisation experiments mentioned above. The low-throughput tab contains single-cell series that were performed using the Fluidigm C1 System.

## Series datasets
A Series is a grouping of datasets that share a biological or functional theme. A dedicated landing page for the most popular series was developed to help explore them easily and is subdivided into seven different tabs. The link for this page can be found here: http://tinyurl.com/ENCODE-Series. These tabs include organism development, treatment time, treatment concentration, replication timing, gene silencing, differentiation, and disease. Other notable series that are not displayed on this page are the reference epigenomes series (http://tinyurl.com/ReferenceEpigenome) and multiOmic series (http://tinyurl.com/MultiomicsSeries). More detailed descriptions about all

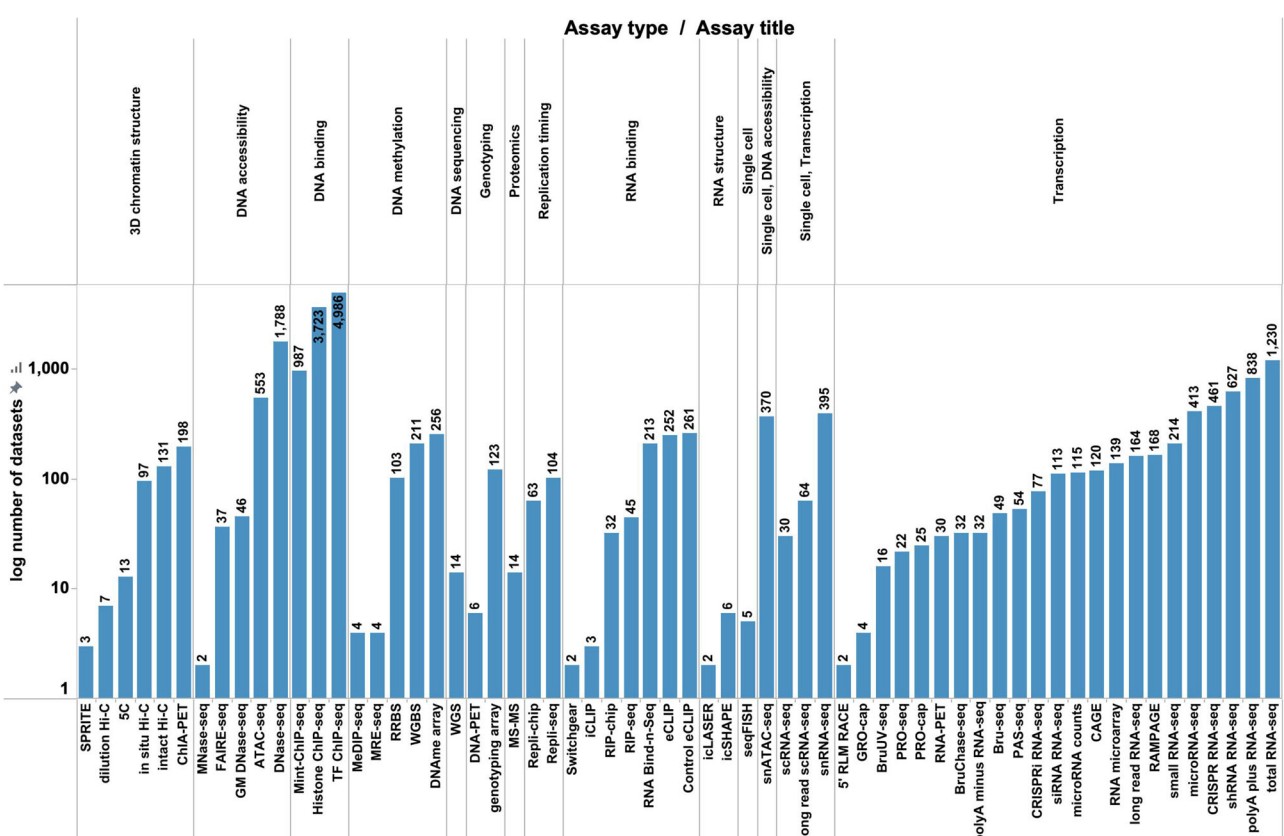

**Fig. 2 | ENCODE functional genomics experiments classified by assay types.** The figure provides a complete breakdown of all the ENCODE functional genomics experiments classified by different assay types. As evident from the plot, the maximum diversity of ENCODE assays is found within the Transcription based assays which included 23 different assay types. In addition to transcription based assays, the various other assay types include 3D chromatin structure, DNA accessibility, DNA binding, DNA methylation, DNA sequencing, Genotyping, Proteomics, Replication timing, RNA binding, RNA structure, single cell (general), Single cell (DNA accessibility) as well as Single cell (Transcription).

the various series can be found in the Supplemental Information and Table 1.

## Computational and integrative products (Annotations)

With 69,703 annotation datasets (Fig. 4) currently on the portal, these make up the largest number of datasets found on the ENCODE website (http://tinyurl.com/ENCODE-Annotations). Annotation datasets are home to data generated in different ENCYCLOPAEDIA[17] versions, including current, v4, v3, v2, v1, v0.3 and v0.2, as well as several other computationally predicted genome annotations derived using integrative analysis of several processed output files from functional genomics assays. A large proportion of the community datasets submitted to the portal host files that are utilised in a sister project called RegulomeDB[18] that has calculated the functional significance of SNPs (https://regulomedb.org/regulome-search/).

A special Encyclopaedia browser was developed to visualise the various annotations on a genome browser along with the ability to select specific tracks of interest (http://tinyurl.com/encyclopaedia-browser). This page is divided into two tabs - one tab for humans and the other for mouse, both of which display the various encyclopaedia annotations from different biosamples. A body map-based filtering is available on both tabs and allows one to filter the tracks for specific organs or biosamples of interest. In addition, the tracks can also be filtered down using the biochemical activity facet to display only one (or any) of CTCF, DNase-seq, H3K27ac or H3K4me3 derived annotations. The browser tracks can also be sorted either by biosample term name or annotation type using the sort buttons. Navigation to specific gene locations can be performed using the search by gene box. An ability to navigate to specific chromosome locations (using the

genome coordinates) is also available on the topmost section of the browser. The cCRE tracks are coloured to display the predicted biochemical activity, and the legends of each can be found on the top right corner of the genome browser.

## ENCODE Collections

Collection pages are dedicated interfaces that display specific sets of functional genomic experiments. We have a total of 11 collection pages (Fig. 5 and Table 2). Based on data obtained using Google Analytics, between 1st Jan 2022 and 31st Dec 2022, the five top-most visited collection pages include the ENCORE matrix, ENTEx matrix, Rush Alzheimer's Disease Study matrix, Mouse Development matrix and Deeply Profiled Cell Line matrix. For more detailed description of all the 11 collections, refer to Supplemental data.

## New home page

To provide an enhanced experience, we have completely remodelled the ENCODE Portal home page. The home page now provides a way to navigate through the wide variety of data types and collections available on the data portal. As demonstrated in Fig. 6, the top section of the page contains a search box that can be used to find objects relating to the search input (see below for more details). Below the search box, there are several clickable "cards" with a small logo, name, and description (which can be seen by selecting the question mark at the top left corner of the card). Upon selecting each of the individual cards, the respective landing pages for each can be viewed for further data exploration.

The cards on the homepage are laid out in four different panels. The top two panels (orange boxed cards in Fig. 6) interact with the search box while the bottom two (green boxed cards in Fig. 6) do not.

| Organism | Biosample classification | Elements selection method | CRISPR screen | | | | Massively parallel reporter assay | | Single cell | |
|---|---|---|---|---|---|---|---|---|---|---|
| | | | CRISPR screen | FACS CRISPR screen | Flow-FISH CRISPR screen | proliferation CRISPR screen | MPRA | STARR-seq | perturbation followed by scRNA-seq | perturbation followed by snATAC-seq |
| *Homo sapiens* | cell line | DHS (DNase hypersensitive sites) | | | 240 | 10 | 19 | | | |
| | | DHS, transcription start sites, chromatin states | | | | | | 2 | | |
| | | accessible genome regions | | | | 6 | | 2 | | |
| | | histone modifications | | | | 6 | | | | |
| | | none | 3 | 8 | 40 | 25 | 28 | 24 | 2 | 8 |
| | | sequence variants | | | | | 48 | | | |
| | | synthetic elements | | | | | 1 | | | |
| | | transcription factor binding sites | | | | | 10 | | | |
| | | transcription start sites | | | | | | 1 | | |
| | in vitro differentiated cells | none | | 6 | | 2 | 2 | | | |
| | organoid | accessible genome regions, histone modifications | | | | | | 2 | | |
| | | accessible genome regions, histone modifications, DHS | | | | | | 2 | | |
| | primary cell | none | | | | | 3 | | | |
| | | sequence variants | | | | | 1 | | | |
| *Mus musculus* | cell line | DHS (DNase hypersensitive sites) | | | 5 | | | | | |
| | | none | | | 5 | | 6 | | | |
| | primary cell | accessible genome regions | | | | | | 6 | | |

**Fig. 3 | Overview of the ENCODE functional characterisation experiments.** The figure demonstrates the diversity of ENCODE functional characterisation experiments classified by organism, biosample classification, element selection method and various assays. As evident in the figure, the maximum number of functional characterisation experiments available on the ENCODE portal were performed on *Homo sapiens* derived cell lines which were assayed using CRISPR screen based assay type. In addition, several reporter based assays are also available from cell line samples originated from *Homo sapiens*. Moreover, some reporter assays were also performed on human organoids and primary cells. A handful of Mus musculus based functional characterisation experiments are also available from primary cells and cell lines.

## Table 1 | Table summarising all the ENCODE series

| Series | Description | Total numbers | Link |
|---|---|---|---|
| **Organism Development series** | Set of related functional genomics datasets performed to study the changes that occur within the genomic landscape of a particular biosample type during the developmental stages of an organism. | 433 | https://tinyurl.com/Organism-DevelopmentSeries |
| **Treatment time series** | Set of functional genomics assays that were performed on the same biosample but sampled at different treatment durations | 82 | http://tinyurl.com/TreatmentTimeSeries |
| **Treatment concentration series** | Set of functional genomics datasets that were assayed on a particular biosample for a fixed time but treated with different concentrations of the same chemical. | 2 | http://tinyurl.com/TreatmentConcentrationSeries |
| **Replication timing series** | Set of functional genomics datasets that assayed different primary cells and cell lines sampled at various cell cycle phases, including early S, late S, G1b, S1, S2, S3, S4 and G2. | 22 | http://tinyurl.com/ReplicationTimingSeries |
| **Gene silencing series** | Two types: (A) Pair of functional genomics datasets performed on biosamples that contain either a normal copy or a silenced copy of a target gene using either siRNA, shRNA, or CRISPR editing (B) Pair of control and experimental datasets that are depleted of a particular target protein using an auxin-inducible degron (AID) method. | 1196 | http://tinyurl.com/GeneSilencingSeries |
| **Differentiation series** | Set of related functional genomics datasets wherein assays were performed on in vitro differentiated cells or organoids that were derived from embryonic stem cells (such as H9) or pluripotent stem cells (such as GM23338 and WTC11). | 41 | http://tinyurl.com/Differentiation-Series |
| **Disease series** | Two types: (A) Groups assays performed on brain samples that have been collected from the Rush Institute from patients diagnosed with different levels of cognitive impairments, and (B) Pair of assays performed on tissues derived from non-cancerous and cancerous samples from the same patient. | 132 | http://tinyurl.com/DiseaseSeries |
| **Reference epigenome series** | Collection of datasets profiling all the epigenomic marks of a particular biosample, ideally collected from the same individual donor. | 325 | https://tinyurl.com/ReferenceEpigenomeSeries |
| **Multiomics series** | Set of paired multiomics assays (snATAC-seq and snRNA-seq) performed on a particular biosample. | 160 | https://tinyurl.com/encode-multiomics-series |

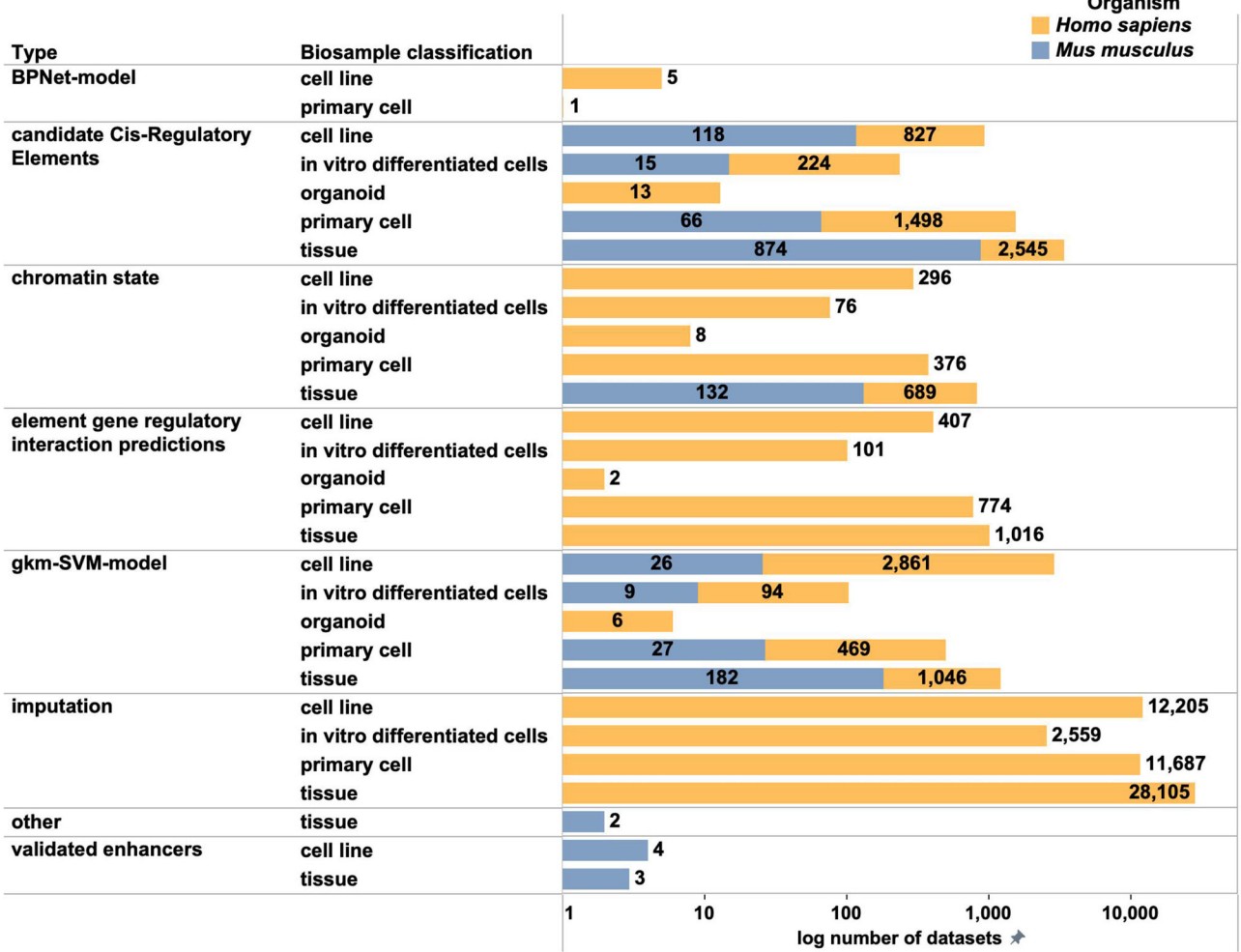

**Fig. 4 | ENCODE computational and integrative products (Annotations).**
The figure demonstrates the total numbers of available computational and integrative products (annotation datasets) classified by organism, biosample classification and annotation type. As evident from the figure, the maximum number of annotations are available within the Imputation category derived from *Homo sapiens* samples. The candidate Cis-Regulatory elements (cCREs) form the basis of the various ENCYCLOPAEDIA[17] versions.

The topmost panel consists of the three major ENCODE data-type cards: Functional genomics experiments, Functional characterisation experiments and Encyclopaedia of elements (specifically representative DHSs, conserved cis-Regulatory Elements; cCREs, and chromatin state models). All the cards below the top panel consist of useful subsets of the main data corpus organised on different pages (see the above section on Collection pages). The functional genomics experiments card links to a matrix view displaying the various assays of this category, organised by different biosamples. The functional characterisation experiments card also links to a matrix view that displays all the assays classified by organism and biosamples. Similarly, the Encyclopaedia of elements card links to a search results page hosting the current version of the ENCODE encyclopaedia, including candidate *cis*-regulatory elements (cCREs), chromatin states and representative DNase hypersensitivity sites.

The cards located on the second panel (grey coloured cards) include combinations of several ENCODE collections (more details above), such as the Rush Alzheimer's disease study matrix, ENTEx matrix, deeply profiled cell lines matrix, human donors matrix, ENCORE matrix and the stem cell development page. In addition, we have cards that link to the Computational integrative products matrix and a search page for Imputed datasets. The third panel (red coloured cards) consists of cards that link to other collections, series and applications that do not interact with the search box.

The collections include Immune cells matrix, Mouse development matrix and Reference epigenomes matrix. The series cards include the Functional genomics series search and the Single cell experiments search. In addition, we have the RNA-get, Region-search, Encyclopaedia browser and ChIP-seq experiments matrix. The fourth panel (blue coloured cards) links to the various helpful pages such as Materials and methods, Publications and Getting-started.

### Search box usage

The search box on the top-most part of the home page can be used to search either the ENCODE portal or the SCREEN[17] site (https://screen.wenglab.org). A toggle button is provided to select one of the two options. SCREEN is a tool that was developed to easily browse and search the candidate cis-Regulatory elements predicted by using the entire ENCODE data corpus.

For example, as shown in Fig. 7, typing "H3K4me3" in the search box, and moving the toggle to ENCODE, the search for "H3K4me3" matches all cards that include any datasets containing this search term and are highlighted in yellow. In addition, a smaller box below the search box will appear that displays a list of all the different object-types on the ENCODE portal that contain search results related to H3K4me3. Each of the items in the objects list is clickable and redirects to the corresponding page for that object type, along with that search

**Assay type**

| Collection | Organism | 3D chromatin structure | DNA accessibility | DNA binding, histone | DNA binding, TF | DNA methylation | DNA sequencing | Genotyping | Proteomics | Replication timing | RNA binding | Single cell | Single cell, DNA accessibility | Single cell, Transcription | Transcription | Total datasets |
|---|---|---|---|---|---|---|---|---|---|---|---|---|---|---|---|---|
| Deeply profiled cell lines | *Homo sapiens* | 30 | 13 | | | | | | | | | | 15 | | 133 | 191 |
| Degron | *Homo sapiens* | 45 | 31 | 92 | 16 | | | | | | | | 6 | | 62 | 252 |
| ENCORE | *Synthetic constructs* | | | | | | | | | | 213 | | | | | 213 |
| | *Homo sapiens* | | | | | 68 | | | | | 516 | | | | 1,009 | 1,593 |
| ENTEx | *Homo sapiens* | 8 | 119 | 441 | 242 | 127 | 12 | 4 | | | 4 | | 45 | | 276 | 1,278 |
| H1 stem cells | *Homo sapiens* | 4 | 8 | 136 | 86 | 8 | | 1 | 3 | 1 | 1 | | | | 64 | 312 |
| H9 stem cells | *Homo sapiens* | 3 | 20 | 201 | 23 | 8 | | | 1 | | 8 | | 8 | | 41 | 313 |
| Human Donor | *Homo sapiens* | 174 | 1,444 | 2,746 | 456 | 288 | 12 | 44 | | 17 | 2 | 5 | 273 | 138 | 1,379 | 6,978 |
| Human reference epigenomes | *Homo sapiens* | | 287 | 2,056 | 283 | 91 | | | | | | | | | 342 | 3,059 |
| Immune cells | *Homo sapiens* | 64 | 509 | 948 | 15 | 7 | | | | 1 | | | 28 | 1 | 163 | 1,736 |
| Mouse reference epigenomes | *Mus musculus* | | 108 | 669 | 62 | 72 | | | | | | | | | 175 | 1,086 |
| PGP stem cells | *Homo sapiens* | 3 | 10 | 73 | 29 | 5 | 2 | 1 | | | | | 4 | | 29 | 156 |
| Rush Alzheimers Disease Study | *Homo sapiens* | 3 | 190 | 176 | 59 | | | | | | | 5 | | | 172 | 605 |

**Fig. 5 | ENCODE collections.** The figure provides a breakdown of the numbers of functional genomics experiments represented within various ENCODE collections classified by assay type and organism. As evident from the figure, the maximum number of assays can be found within the Human Donor matrix followed by the Human reference epigenomes matrix. Each collection has its own specific importance and more details about the same can be found in the Supplemental information. *Note: the numbers reflected in this diagram exclude control ChIP-seq datasets.*

## Table 2 | Table summarising all the ENCODE collections

| Collection | Description | Total numbers* | Link |
|---|---|---|---|
| **ENCORE matrix** | The ENCORE matrix summarises assays performed to study protein-RNA interactions of RNA-binding proteins (RBPs) encoded in the human genome. | 1832 | http://tinyurl.com/encore-matrix |
| **Immune cells matrix** | Immune cells matrix summarises epigenomic profiling of human immune cells at different cellular fates and states, including activation, stimulation, and disease (MS). | 1736 | http://tinyurl.com/Immune-matrix |
| **Stem cells differentiation matrix** | Stem cells differentiation matrix summarises assays performed on 3 different stem cells (H1, H9, PGP) and their in vitro differentiated derivatives. | 781 | http://tinyurl.com/stem-cell-matrix |
| **Deeply profiled cell lines matrix** | The deeply profiled cell lines matrix represents data from cell lines that have been extensively studied within the ENCODE project. | 6474 | http://tinyurl.com/deeply-profiled |
| **Protein Knockdown/Degron matrix** | This matrix displays experiments that were performed on transformed HCT116 cell lines, utilising the auxin-inducible Degron (AID) system. | 252 | http://tinyurl.com/degron-matrix |
| **Human donor matrix** | The Human Donor Matrix provides a matrix view of assays performed on biosamples derived from individual human donors. | 6978 | http://tinyurl.com/human-donor-matrix |
| **ENTEx matrix** | The ENTEx matrix displays approximately 30 overlapping tissues from four individual donors, which were sampled in a collaborative effort with the GTEx Consortium. | 1567 | http://tinyurl.com/entex-matrix |
| **Rush Alzheimer's Disease Study Matrix** | Human brain samples collection page that includes data from multiple brain regions collected from individuals with various levels of cognitive impairment. | 664 | http://tinyurl.com/brain-matrix |
| **Reference Epigenomes matrix** | Project data from human tissue, cell line, primary cell, and in vitro differentiated cell biosamples, organised as reference epigenomes following guidelines set out by IHEC. | 4145 | http://tinyurl.com/reference-epigenome-matrix |
| **Mouse development matrix** | The mouse development matrix displays embryonic to postnatal mouse developmental time course data across several tissues, organised as reference epigenomes. | 1934 | http://tinyurl.com/mouse-development-matrix |

*Note: Refer to Fig. 5 for a more comprehensive breakdown of the total number of datasets within each category.

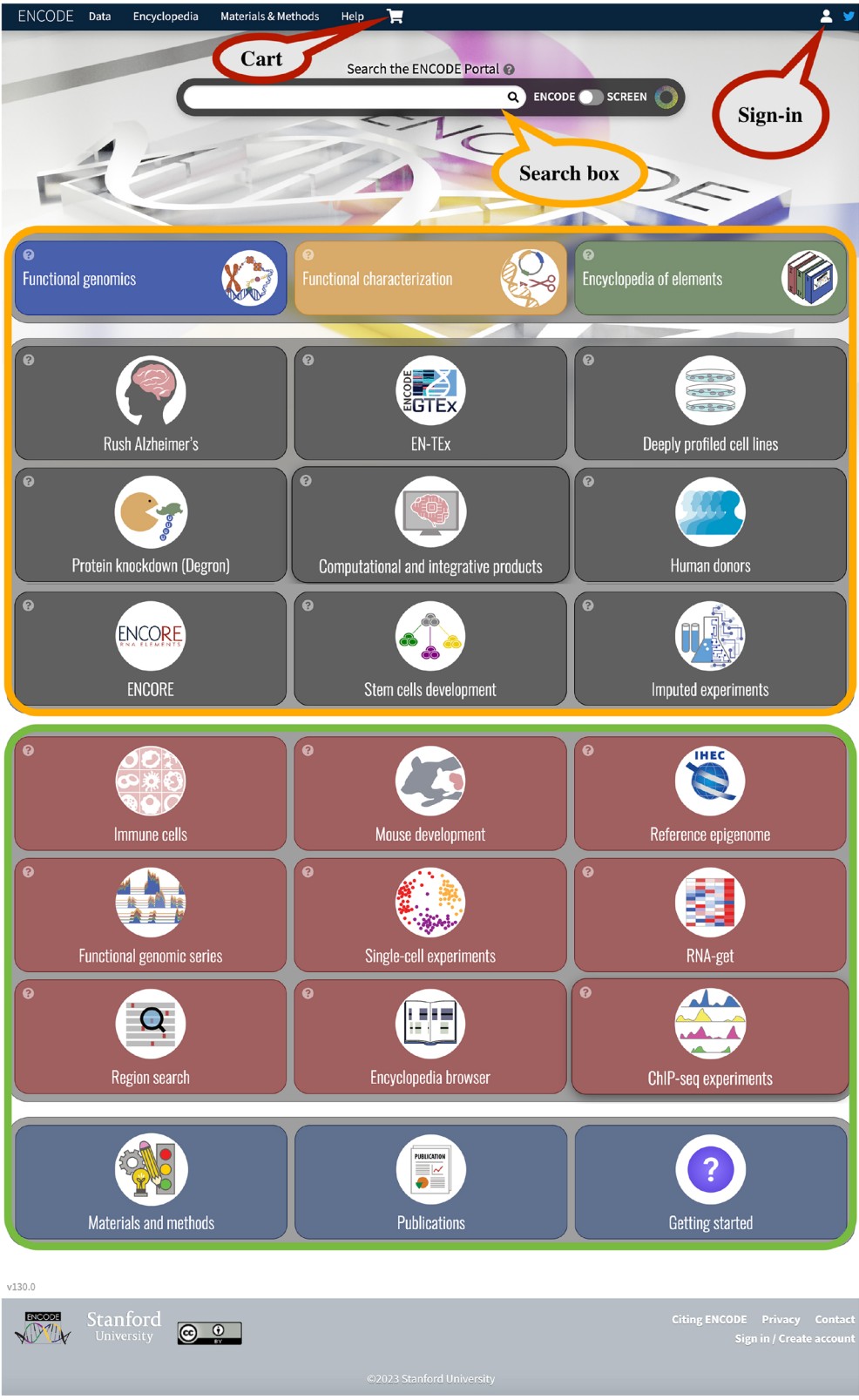

**Fig. 6 | ENCODE home page.** The figure demonstrates the various features of the ENCODE home page. The home page displays several clickable cards which provide shortcut links to different ENCODE datasets, collections as well as links to help pages. There is also a prominent search box feature which helps users type in keywords and search within the ENCODE datasets as well as the SCREEN registry. The home page also allows users to navigate different functionalities such as exploring the Cart features and the Sign-in options.

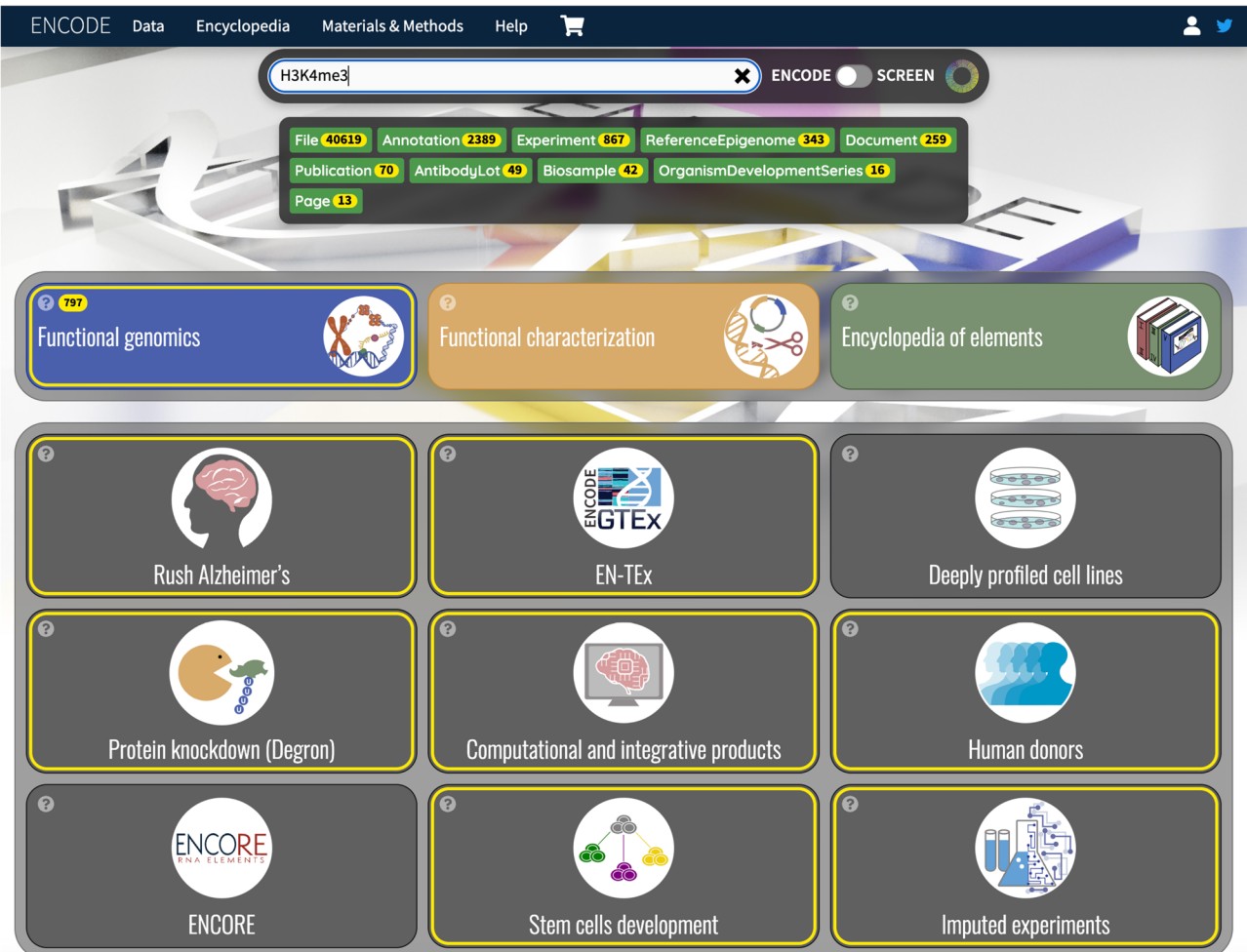

**Fig. 7 | ENCODE Search box usage example.** This figure demonstrates what a user might see when trying to search "H3K4me3". As seen in figure, when a user hits enter after typing the keyword, the search box highlights relevant homepage cards that have any data relevant to the keyword search term. In this case, 8 out of 12 cards are highlighted yellow, indicating the presence of H3K4me3 relevant data within those respective cards. The cards that are not highlighted have no metadata related to H3K4me3. In addition to the yellow highlights below the cards, notice the black box section listing the different object types below the search box along with a number indicating the number of specific objects having the keyword. In this example, we can notice that there are 40619 Files and 2389 Annotations that have data relevant to H3K4me3.

term appended in the query. For example, clicking on File will lead to https://www.encodeproject.org/search/?type=File&searchTerm=H3K4me3.

In contrast, when searching the SCREEN database, an option to select either human or mouse genes is shown. For example, searching for CTCF and clicking on the box "Human GRCh38" opens a new page that links to the following page: https://screen.wenglab.org/search/?q=CTCF&uuid=0&assembly=GRCh38. Once on this page, one can navigate to various tabs showing different results for CTCF from the SCREEN registry.

### Carts − a collection of user defined datasets

Carts allow users to select custom combinations of datasets, including functional genomics, functional characterisation, or computational annotations. A user can also add series (sets of datasets) to a cart, effectively adding all the member datasets from that series. Every dataset search page contains a cart icon next to the individual search results when using the search pages. Clicking on those cart icons adds the individual dataset to the cart one at a time. In addition, every search page contains a button on the top section, which is named "Add all datasets to cart". Clicking on this button will allow many datasets to be added to the cart at the same time. A single cart can contain a maximum total of 8000 datasets, and each user is allowed 30 different carts at a given time. The total number of datasets added to a cart can be seen on the top menu bar under the cart icon. The carts feature is available only when a user is logged in. (Disclaimer: to login, users must register an email address with the ENCODE portal; there is no charge for using it and we do not share this email).

The different carts owned by a single user can be managed using the cart-manager page https://www.encodeproject.org/cart-manager/. This page lists all the existing carts and allows them to be annotated by adding a useful title and a short description. Each cart also has a unique identifier and a unique URL, hence, this cart URL references this custom collection and can be shared in publications or other documentation as needed. In addition, there is also an option of making your cart "Listed", which will allow a released cart to be viewed publicly on https://www.encodeproject.org/search/?type=Cart&status=listed&status=released. There is a reversible "lock" to prevent the cart from being altered. Renaming and deleting of existing carts is also allowed from the cart manager. There are two main sections on the cart pages: Different tabs on the page allow one to view the files in various ways. Filters on the left-hand side allow to further sub-select files of interest. The carts allow visualisation of genomic tracks from an arbitrary set of datasets on the Valis browser and to select subsets of

files to download (rather than downloading by search result or individual dataset). To further explore all the cart features, refer to the video tutorials listed here: https://www.encodeproject.org/help/cart/.

## Discussion

The ENCODE portal serves as an enormous data repository for functional genomics experiments. The data has been collected by performing experiments on many legacy cell lines and a wide array of tissue and cell samples obtained from human and mouse. This diverse and massive warehouse of data − the majority of which was processed uniformly − is of unparalleled value to the research community and will greatly increase our understanding of the human genome and the basis for human variation and disease. Researchers can use this data by either comparing their own lab generated experimental data to the ENCODE data or build upon existing knowledge bases, creating novel hypotheses for future experiments. For the ease of exploring the entire corpus of ENCODE and the resultant output files, we have utilised state-of-the-art methods to develop a very dynamic user interface. The ability for a user to create custom data carts and visualise them in the integrated data browser has proven to be a powerful method for integrating results across several data modalities. For the past decade, this resource has been heavily used by researchers worldwide, and even though the project has officially ended, we strongly believe that this resource will remain relevant for many decades to come.

### Ethics statement

The ENCODE database contains data from experiments on human subjects and mice. All human biosamples were collected with open access consent that met relevant IRB standards. All mouse biosamples were approved by the respective institutional animal care and use committees. Each contributing laboratory has provided Institutional Certificates to NHGRI that state their adherence to these policies. All human genetic and genomic data from the ENCODE project is consented for public release without restriction. The ENCODE Consortium members are required to adhere to the open access consent policies defined here: https://www.genome.gov/Funded-Programs-Projects/ENCODE-Project-ENCyclopedia-Of-DNA-Elements#informed-consent.

### Reporting summary

Further information on research design is available in the Nature Portfolio Reporting Summary linked to this article.

## Data availability

All the data discussed throughout this manuscript is openly available on https://www.encodeproject.org. Relevant links are provided throughout the manuscript as needed. Source data for all the figures are provided in this paper. Useful tutorials can be found on https://www.youtube.com/@encodeportal8658.

## Code availability

The ENCODE codebase is available on https://github.com/ENCODE-DCC/encoded The different ENCODE uniform processing pipelines are available at: https://github.com/ENCODE-DCC/atac-seq-pipeline, https://github.com/ENCODE-DCC/chip-seq-pipeline2, https://github.com/ENCODE-DCC/hic-pipeline, https://github.com/ENCODE-DCC/dnase-seq-pipeline, https://github.com/ENCODE-DCC/long-read-rna-pipeline, https://github.com/ENCODE-DCC/rna-seq-pipeline, https://github.com/ENCODE-DCC/wgbs-pipeline, https://github.com/ENCODE-DCC/mirna-seq-pipeline.

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

## Acknowledgements

All the figures in the manuscript are prepared using the free academic licence version of Tableau Desktop 2023.1. The authors would like to thank Salesforce for providing us with a free academic version. In addition, we would like to thank all the past and present members of the ENCODE and affiliated consortia. National Human Genome Research Institute, National Institutes of Health [U24 HG009397]. The content of this manuscript is solely based on the author's views and does not represent the views of the National Human Genome Research Institute or the National Institutes of Health. The funding agencies had no contribution in the decision to publish, nor had any role in the preparation of the manuscript.

## Author contributions

All the figures, the main text, as well as supplemental data of this manuscript was prepared by M.S.K. B.C.H., I.G., J.J. and J.M.C. provided suggestions and editing for the manuscript. B.C.H., C.A.S., I.G. and J.M.C. provided managerial support and leadership. B.L., C.S., C.A.S., I.W., I.G., I.Y., J.S.S., J.H., J.J., J.N.A., K.A., K.Z.L., M.S.K., P.S., Y.L. and Z.M. did all of the data wrangling, database schema design and validation, content management and quality control for the website and database. ES, FT, and PAdenekan wrote the user interface. K.G., P.S., J.S.S., J.-W.L., O.J. and PAssis maintained and extended the backend database

software and data processing code. C.L., M.S., O.J., M.S., S.M. and W.Z. provided system administration and DevOps support for the project.

## Competing interests

The authors have no competing interests.
