## [Peer Review file · Nature Communications]

Data navigation on the ENCODE Portal

Corresponding Author: Dr Benjamin Hitz

Version 0:

Reviewer comments:

Reviewer #1

(Remarks to the Author)

The manuscript describes the most recent updates to the ENCODE web portal, going through available datasets and newly implemented features. The ENCODE portal is amazing resource and the manuscript provides a good summary of its content and of some of its new features.

That being said, some changes are needed to the structure of the manuscript to really help the reader navigate through the content. In short, a description of what will be described (or not described) is needed in the intro. Also, some of the content should be moved to supplements and a conclusion/discussion is needed.

Major comments:

1) I found some of the detailed information to be a bit too much. Perhaps some of the content could be moved to a supplemental text? Examples of things that I think could go to supplements:

- 7 example of series datasets. Only a high-level description could be in the main text.

- Detailed ENCODE collections, same comment. I think only the high-level description + Fig 5 should be in the main text.

2) There is a large amount of exhaustive lists and counts across the manuscript could distract the reader's attention, and are not that interesting to read in the context of a paper. It would make for a lighter read to keep lists short and only a total amount for all items, as anyway those numbers and lists could change rapidly with time. One such example at line 450: "The matrix is subdivided into five major immune cell types along the y-axis: T cells (37 different cell types), B cells (7 different cell types), myeloid cells (8 different cell types), NK cells (2 different cell types) and mononuclear cells (2 different cell types). The x-axis represents the various assay types including: different flavours of RNA-seq, snATAC-seq, ATAC-seq, DNase-seq, various flavours of ChIP-seq (TF-ChIP, Histone ChIP and MINT-ChIP), ChIA-PET, WGBS, intact HiC, Repli-Chip."

3) The manuscript ends with a description of CARTS (which I understand) and also RNA-Get, less clear why, and a long glossary. You need to improve the structure of the paper with a summary at the beginning of what you will describe.

4) There is also no conclusion/discussion. Some conclusion on outlook for the portal is also needed.

Minor comments:

Line 28: "web portal" rather than "web-portal"?

Line 79: The concept of Object should be defined in a broad sense in the beginning. What does it mean to have 2.5 millions objects? It can be understood from the context by reading further, but defining it at first would make for an easier read.

Line 84: There are seven main sub-categories, but only 6 are enumerated.

Line 105 "The facets also serve [..]" Sentence is a bit convoluted and should be split.

Page 6, line 113: Not sure about the format of these notes. Maybe they could be added as supplements?

Page 9, line 170: Some of these links are cumbersome in the text, not sure about the formatting.

Line 221: How can they be accessed?

Page 20: You jumped to Fig 5, I didn't see a description of Fig 4 in the main text.

Page 29, new hope page: Maybe it's just me but on the website it wasn't clear what were the various large groups of cards. What are all the cards in grey? The cards in red, etc. Maybe you need to add sub-headings?

Line 469: The grid with the categories explorer is really nice. There might one tiny issue, unless I misunderstood something:

- Open the link provided
- Click on Hide All
- Click on mesenchymal stem cell in the graph, to enable only this category
- Click on column header H4K91ac
- A result page will list experiments regardless of cell type (instead of just mesenchymal stem cells)

Line 731: The glossary should be moved to the supplemental section.

Reviewer #2

(Remarks to the Author)

- What are the noteworthy results?
- Will the work be of significance to the field and related fields? How does it compare to the established literature? If the work is not original, please provide relevant references.

Kagda et al. describe the ENCODE data portal. This is useful resource for many researchers working in the (human) genomics field. The different aspects of the portal are extensively described.

- Does the work support the conclusions and claims, or is additional evidence needed?
- Are there any flaws in the data analysis, interpretation and conclusions? - Do these prohibit publication or require revision?
- Is the methodology sound? Does the work meet the expected standards in your field?
- Is there enough detail provided in the methods for the work to be reproduced?

There is no new data in this paper and no claims are made so with regard to the questions regarding the "support [of] the conclusion and claims" or "flaws in the data analysis", "methodology" or "enough detail provided in the methods" I would have to answer that no useful comments can be made.

Reviewer #3

(Remarks to the Author)

The manuscript entitled "Data Navigation on the ENCODE Portal" by Kagda et al. is exceptionally well-crafted, offering a clear and comprehensive exposition of the available data within the ENCODE portal, along with instructive insights on how to efficiently explore data of interest through the user-friendly graphical user interface (GUI).

The reviewer acknowledges the manuscript's excellence and offers minor suggestions for refinement:

- 1) The introductory paragraph on page 5 delineates metadata objects and enumerates their total count at 2,534,857. While the total count of file objects is thoughtfully provided, a comprehensive breakdown of other object types is noticeably absent. It would be advantageous to incorporate a TABLE that meticulously outlines the distribution of all objects and categories.
- 2) In Line 85, the phrase "The seven main sub-categories...". A clarification is warranted here; should it be six sub-categories instead?
- 3) On Line 105, the phrase "downsizing of the search results" is used. Considering that the interface adeptly accommodates the inclusion of new facet terms, which might actually expand the search results, a more apt phrase like "refining of the search results" might be more fitting.
- 4) Line 154 discusses how experiments "may either host only...". To enhance precision, the word "typically" could be included to read as "typically either host only".
- 5) On Line 508, the phrase "batch growth are : A673, ...", remove the white space after the colon, yielding "batch growth are: A673, ...".

Version 1:

Reviewer comments:

Reviewer #1

(Remarks to the Author)

The authors have carefully addressed all my comments. I found the manuscript to be much improved.

Minor point: you still refer to the "web-portal" in the abstract instead of "data portal".

Reviewer #2

(Remarks to the Author)

I have no further comments.

Reviewer #3

(Remarks to the Author)

The revised manuscript on the ENCODE portal by Hitz et al. has appropriately addressed the reviewers' feedback and comments. I have no further advice except for one minor suggestion: in line 5 of the abstract, change "web-portal" to "data portal" to maintain consistency throughout the paper.

REVIEWER COMMENTS

Reviewer #1 (Remarks to the Author):

The manuscript describes the most recent updates to the ENCODE web portal, going through available datasets and newly implemented features. The ENCODE portal is an amazing resource and the manuscript provides a good summary of its content and of some of its new features.

That being said, some changes are needed to the structure of the manuscript to really help the reader navigate through the content. In short, a description of what will be described (or not described) is needed in the intro. Also, some of the content should be moved to supplements and a conclusion/discussion is needed.

Changes in the introduction: Edited the last paragraph to include “what will be described” in the introduction (new lines 70-74). Original lines 69-77 deleted.

Added a conclusion (new lines 325-341).

Added two new tables Table 1 (summarizing all the ENCODE series) and Table 2 (Table summarizing all the ENCODE collections).

Moved the Notes (original lines 113-118 to Supplemental data lines 360-366), ENCODE series (original lines 222-391 to Supplemental data lines 3-165), ENCODE collections (original lines 419-614 to Supplemental data lines 167-377), Glossary (Supplemental data lines 368-572) and RNA-get (Supplemental data lines 338-359) sections to supplemental data.

Changed the Notes (original lines 145 to 150 into a paragraph correspond to new lines 130-136)

Edited lengthy links to tinyurl links which are also used in the Tables 1 and 2.

Major comments:

1) I found some of the detailed information to be a bit too much. Perhaps some of the content could be moved to a supplemental text? Examples of things that I think could go to supplements:

- 7 example of series datasets. Only a high-level description could be in the main text.

- Detailed ENCODE collections, same comment. I think only the high-level description + Fig 5 should be in the main text.

Moved the detailed description of series and collections to supplemental data. ENCODE series (original lines 222-391 to Supplemental data lines 3-165), ENCODE collections (original lines 419-614 to Supplemental data lines 167-377)

2) There is a large amount of exhaustive lists and counts across the manuscript could distract the reader's attention, and are not that interesting to read in the context of a paper. It would make for a lighter read to keep lists short and only a total amount for all items, as anyway those numbers and lists could change rapidly with time. One such example at line 450:

"The matrix is subdivided into five major immune cell types along the y-axis: T cells (37 different cell types), B cells (7 different cell types), myeloid cells (8 different cell types), NK cells (2 different cell types) and mononuclear cells (2 different cell types). The x-axis represents the various assay types including: different flavours of RNA-seq, snATAC-seq, ATAC-seq, DNase-seq, various flavours of CHIP-seq (TF-CHIP, Histone CHIP and MINT-CHIP), ChIA-PET, WGBS, intact HiC, Repli-Chip."

Removed long lists and numbers throughout the main manuscript as relevant.

3) The manuscript ends with a description of CARTS (which I understand) and also RNA-Get, less clear why, and a long glossary. You need to improve the structure of the paper with a summary at the beginning of what you will describe.

Moved the glossary and RNA-get to supplemental data. Glossary (Supplemental data lines 368-572) and RNA-get (Supplemental data lines 338- 359)

4) There is also no conclusion/discussion. Some conclusion on outlook for the portal is also needed.

Added a discussion section (new lines 325-341).

Minor comments:

Line 28: "web portal" rather than "web-portal"? Changed to "data portal": new line 28

Line 79: The concept of Object should be defined in a broad sense in the beginning. What does it mean to have 2.5 millions objects? It can be understood from the context by reading further, but defining it at first would make for an easier read.

Removed lines 79-84 and shortened the first paragraph under "Summary of Encode datasets and objects" (new lines 77-82)

Line 84: There are seven main sub-categories, but only 6 are enumerated. Fixed (new line 78)

Line 105 "The facets also serve [..]" Sentence is a bit convoluted and should be split.

Edited (New lines 97-100): "The facets also serve as important pointers about the variety of data available for selection. The facets have been grouped into relevant categories such that each of these grouped facets could be expanded or collapsed as needed."

Page 6, line 113: Not sure about the format of these notes. Maybe they could be added as supplements?

Moved to supplemental data. (original lines 113-118 to Supplemental data lines 360-366)

Page 9, line 170: Some of these links are cumbersome in the text, not sure about the formatting.

Created tiny urls to avoid long links at new line 157 (<http://tinyurl.com/Functional-characterization>).

Line 221: How can they be accessed? Provided tiny url links at new lines 197,198 (<http://tinyurl.com/ReferenceEpigenome>) (<http://tinyurl.com/MultiomicsSeries>).

Page 20: You jumped to Fig 5, I didn't see a description of Fig 4 in the main text.
Added at new line 201

Page 29, new home page: Maybe it's just me but on the website it wasn't clear what were the various large groups of cards. What are all the cards in grey? The cards in red, etc. Maybe you need to add sub-headings?

Answer: The cards in grey (highlighted orange in Fig 6) interact with the search box, while the cards in red and blue (highlighted green in Fig 6) DO NOT interact with the search box. There isn't a very clear demarcation (except that one set interacts with the search box while the other doesn't) between types of pages these red and grey cards link to; i.e. they both link to either collections/series of datasets. While the cards in blue link to help related pages.

Line 469: The grid with the categories explorer is really nice. There might one tiny issue, unless I misunderstood something:

- Open the link provided
- Click on Hide All
- Click on mesenchymal stem cell in the graph, to enable only this category
- Click on column header H4K91ac
- A result page will list experiments regardless of cell type (instead of just mesenchymal stem cells)

I think overall you probably got it right except that there is only Repli-ChIP data (highlighted in grey in the screenshot below) for mesenchymal stem

cells in the graph. But yes, you are right, clicking on the column header shows the experiment's search page

(https://www.encodeproject.org/search/?type=Experiment&replicates.library.biosample.donor.accession=ENCDO222AAA&status=released&control_type!=*&assay_title=Histone+ChIP-seq&assay_title=Mint-ChIP-seq&target.label=H4K91ac) with all the stem cells for that corresponding assay regardless of the biosample selected.

If you want to just look at that data for the corresponding cell type and assay type, you will need to click on the grey box within the matrix. In this case it will link you to this url:

https://www.encodeproject.org/search/?type=Experiment&replicates.library.biosample.donor.accession=ENCDO222AAA&status=released&control_type!=*&biosample_ontology.classification=in+vitro+differentiated+cells&biosample_ontology.term.name=mesenchymal+stem+cell&assay_title=Repli-chip

So clicking on rows shows you all the assays performed on that biosample irrespective of the assay type. Clicking on the columns shows you all the assays performed on stem cells irrespective of the biosample type. Clicking

on the shaded box within the matrix narrows down the search results for that particular biosample and that particular assay type.

Line 731: The glossary should be moved to the supplemental section. **Done**

Reviewer #2 (Remarks to the Author):

- What are the noteworthy results?
- Will the work be of significance to the field and related fields? How does it compare to the established literature? If the work is not original, please provide relevant references.

Kagda et al. describe the ENCODE data portal. This is useful resource for many researchers working in the (human) genomics field. The different aspects of the portal are extensively described.

- Does the work support the conclusions and claims, or is additional evidence needed?
- Are there any flaws in the data analysis, interpretation and conclusions? - Do these prohibit publication or require revision?
- Is the methodology sound? Does the work meet the expected standards in your field?
- Is there enough detail provided in the methods for the work to be reproduced?

There is no new data in this paper and no claims are made so with regard to the questions regarding the “support [of] the conclusion and claims” or “flaws in the data analysis”, “methodology” or “enough detail provided in the methods” I would have to answer that no useful comments can be made.

Reviewer #3 (Remarks to the Author):

The manuscript entitled "Data Navigation on the ENCODE Portal" by Kagda et al. is exceptionally well-crafted, offering a clear and comprehensive exposition of the available data within the ENCODE portal, along with instructive insights on how to efficiently explore data of interest through the user-friendly graphical user interface (GUI).

The reviewer acknowledges the manuscript's excellence and offers minor suggestions for refinement:

1) The introductory paragraph on page 5 delineates metadata objects and enumerates their total count at 2,534,857. While the total count of file objects is thoughtfully provided, a comprehensive breakdown of other object types is noticeably absent. It would be advantageous to incorporate a TABLE that meticulously outlines the distribution of all objects and categories.

Answer: Based on the comments from another reviewer, we have decided to eliminate that sentence along with the whole paragraph. We agree those details might be nice to have but considering the length of the paper, we have decided to not include this information.

2) In Line 85, the phrase "The seven main sub-categories...". A clarification is warranted here; should it be six sub-categories instead? **Fixed (new line 78)**

3) On Line 105, the phrase "downsizing of the search results" is used. Considering that the interface adeptly accommodates the inclusion of new facet terms, which might actually expand the search results, a more apt phrase like "refining of the search results" might be more fitting. **Fixed (new line 97)**

4) Line 154 discusses how experiments "may either host only...". To enhance precision, the word "typically" could be included to read as "typically either host only". **Fixed (new line 140)**

5) On Line 508, the phrase "batch growth are : A673, ...", remove the white space after the colon, yielding "batch growth are: A673, ...". **Fixed (Supplemental data line 241)**